# Representing Hyperbolic Space Accurately using Multi-Component Floats

**Tao Yu**
Department of Computer Science
Cornell University
tyu@cs.cornell.edu

**Christopher De Sa**
Department of Computer Science
Cornell University
cdesa@cs.cornell.edu

## Abstract

Hyperbolic space is particularly useful for embedding data with hierarchical structure; however, representing hyperbolic space with ordinary floating-point numbers greatly affects the performance due to its *ineluctable* numerical errors. Simply increasing the precision of floats fails to solve the problem and incurs a high computation cost for simulating greater-than-double-precision floats on hardware such as GPUs, which does not support them. In this paper, we propose a simple, feasible-on-GPUs, and easy-to-understand solution for numerically accurate learning on hyperbolic space. We do this with a new approach to represent hyperbolic space using multi-component floating-point (MCF) in the Poincaré upper-half space model. Theoretically and experimentally we show our model has small numerical error, and on embedding tasks across various datasets, models represented by multi-component floating-points gain more capacity and run significantly faster on GPUs than prior work.

## 1  Introduction

Representation learning is of particular interest nowadays in machine learning. Real-world data is often discrete—such as natural language sentences in the form of syntax trees and graphs [7, 22]—which makes it critical to do a good job of representing and storing them as numbers which we can better understand and deal with. Perhaps the most common way of doing this representation is with *embedding*, in which discrete objects are assigned points in some continuous (typically high-dimensional Euclidean) space. Standard embeddings in NLP include Word2Vec [24] and GloVe [30]. However, many kinds of real-world data are hierarchical or graph-like [25], in which case embedding into non-Euclidean space can outperform the standard Euclidean embedding. *Hyperbolic space* is such an embedding target of particular interest, showing impressive performance on many tasks such as taxonomies, link prediction and inferring concept hierarchies [27, 28, 15].

Hyperbolic space [9] is helpful for many types of embeddings because of its constant negative curvature, as distinguished from Euclidean space with zero curvature. This negative-curvature geometry endows hyperbolic space with many useful properties not present in Euclidean space. For example, the volume of a ball in even two-dimensional hyperbolic space increases exponentially with respect to the radius, as shown in [9], as a contrast to the polynomial rate at which balls grow in Euclidean space. This makes it possible to embed more instances in the hyperbolic ball than Euclidean ball with the same radius, making it ideal for structures like trees where the number of objects at a distance $d$ from any given object can grow exponentially with $d$. Reasoning from this intuition, the hyperbolic space "looks like" a finite-degree tree and can be regarded as a continuous version of trees [6]. The combination of low dimension (which facilitates learning) with high volume (which increases representational capacity) makes hyperbolic space inherently well-suited for embedding hierarchical structures data like trees and graphs.

35th Conference on Neural Information Processing Systems (NeurIPS 2021).

However, the advantages of hyperbolic space do not come for free. The large volumes that are so beneficial from a embedding-capacity perspective create unavoidable numerical problems. Informally called "the NaN problem" [37, 34], representing hyperbolic space with ordinary floating-point numbers can cause significant errors and lead to severe numerical instability that compounds throughout training, causing NaNs to pop out everywhere in the computations. Common models of hyperbolic space used to learn hyperbolic embeddings using floating-point arithmetic, such as the Poincaré ball model [27, 15] and the Lorentz hyperboloid model [28], all suffer from significant numerical errors. These errors originate in the ordinary floating-point representation, and are further amplified in subsequent computations by the very ill-conditioned Riemannian metrics involved in their construction. Yu and De Sa [37] show that in the Lorentz hyperboloid model, the error of representing an exact point with ordinary floating-point numbers becomes unbounded as the point moves away from the origin, which is what will happen in most embedding tasks: points unavoidably move far away from the origin as more and more data instances crowd in.

Many efforts and compromises have been made so as to solve or alleviate the 'NaN' problem. One commonly adopted technical solution firstly exploited in [35] is using an appropriate scaling factor, i.e., scale down all the distances by the scaling factor before embedding, then recover the original distances by dividing to the scaling factor. However, this scaling will increase the distortion of the embedding, and the distortion gets worse as the scale factor increases. Sala et al. [34] show that this sort of tradeoff is unavoidable when embedding in hyperbolic space: one can either choose to increase the number of bits used for the floating-point numbers or increase the dimension of the space, and this is independent of the underlying models. However, increasing the dimension is not a panacea, as the 'NaN' problem still exists in high dimensional space (albeit to a lesser degree).

A straightforward solution to the 'NaN' problem would be to compute with high precision floating-point numbers, such as BigFloats (specially designed with a large quantity of bits), as adopted in the combinatorial construction of hyperbolic embedings in [34], where floating-point numbers with thousands of bits are used with the support of Julia [4]. However, there are two evident problems with representing hyperbolic space using BigFloats. Firstly, BigFloats are not supported on ML accelerator hardware like GPUs: computations using BigFloats on CPUs will be slow and hence not realistic for many meaningful tasks. Secondly, BigFloats are currently not supported in most deep learning frameworks even on CPUs, so one would need to develop the algorithms from scratch so as to learn over hyperbolic space, which is tedious and fraught with the danger of numerical mistakes.

A hybrid solution to the 'NaN' problem proposed by Yu and De Sa [37] is to avoid floating-point arithmetic as much as possible by using integer arithmetic instead—as integer arithmetic can be done exactly. Their *tiling-based* model works by decomposing hyperbolic space into a tiling: the product of a compact fundamental region (which can be represented in floating-point with bounded error) and a discrete group (which supports integer computation). We can also interpret their approach as decomposing a large floating-point number into a multiply of a small floating-point number with a large integer number using some group actions [3, 33]. This allows for a guaranteed bound on the numerical error of learning in hyperbolic space. However, their approach is limited somewhat in that its guarantees apply to loss functions that depend only the hyperbolic distance between points, and their best-performing model (their $L$-tiling model) is limited to 2-dimensional space. A bigger issue with this type of approach is computational: the big-integer matrix multiplication involved in the model is not supported on GPUs and is not yet implemented in many deep learning frameworks such as PyTorch [29] and TensorFlow [1]. This effectively prevents the method from being used in training of complex models that depend on GPU acceleration, such as hyperbolic neural networks [14, 18, 10, 23].

In this paper, we propose a more direct way to **represent and learn in hyperbolic space without sacrificing either accuracy guarantees or GPU acceleration**. To achieve high precision computations and controllable numerical errors, we choose to represent hyperbolic space with multiple-component floating-point numbers (MCF) [31, 32, 19, 36], an alternative approach different from BigFloats where a number is expressed as an unevaluated sum of multiple ordinary floating-point numbers. We make the following contributions:

- We propose representing hyperbolic space with the Poincaré upper-half space model for optimization in section 3, and show how to represent it with MCF in section 4 to eliminate the 'NaN' problem.

- We provide algorithms for computing with these MCF that are adapted to hyperbolic space in section 5 and section 6, where we prove numerical errors can be reduced to any degree by simply increasing the number of components of the MCF.
- We experimentally measure our model on embedding tasks across several datasets, and show that one can gain more learning capacity (compared to models that use ordinary floating-point numbers) with only a mild computational slowdown on GPUs in section 7, meanwhile significantly faster than prior high precision hyperbolic models.

## 2    Related Work

The non-Euclidean geometry of hyperbolic space offers many distinctive properties useful in machine learning. It is well-suited to model hierarchical structures such as trees and more complex graphs: for instance, Sala et al. [34] show that a tree can be embedded into a 2-dimensional hyperbolic space with arbitrarily low distortion, and Gromov [16] shows that any finite tree can be embedded into a finite hyperbolic space such that distances are preserved approximately.

Nickel and Kiela [27] propose to embed symbolic data into the Poincaré ball model of hyperbolic space for learning the hierarchical representations, where the Poincaré ball embeddings outperform Euclidean embeddings significantly on hierarchical data in terms of both representation capacity and generalization ability. Subsequently, Ganea et al. [15] further improve the empirical results of Poincaré ball embeddings on directed acyclic graphs by theoretically grounding entailment using the hierarchical relations. Later work uses other models of hyperbolic space [28, 17] besides the Poincaré ball model for various purposes. For example, the Lorentz hyperboloid model used in Nickel and Kiela [28] to learn continuous hierarchies from large-scale unstructured data, was observed to be substantially more efficient and numerically stable. Along a different direction, hyperbolic embeddings can be used in many down-streaming tasks, by simply treating the hyperbolic embedding as a layer to form neural networks, such as the hyperbolic neural networks [14] for textual entailment and noisy-prefix recognition tasks, hyperbolic attention networks [18] for neural machine translation and visual question answering tasks, and hyperbolic graph networks [10, 23] for link prediction and node classification tasks.

Although widely adopted in many applications, the 'NaN' problem in hyperbolic representations can be observed easily in practice and generally remains unsolved. Sala et al. [34] suggests that hyperbolic embeddings can have high quality on hierarchies but require large dimensions or high precision for graphs with long chains: they suggest alleviating the 'NaN' problem by using scaling factors together with BigFloats. Yu and De Sa [37] introduced tiling-based models, the first theoretical solution to the 'NaN' problem with bounded-everywhere numerical error. The tiling-based models enable deriving high-precision embeddings without using high precision floating-point numbers.

In the literature of arbitrary precision algorithms, there are two different ways to achieve high precision computations, and BigFloat is not the only option. The first one is straightforward: the multiple-digit format, which consists of a sequence of digits coupled with a single exponent. Examples in this category include the computation package Mathematica [20], the BigFloats format in Julia [4], and the Fortran multiple-precision arithmetic packages [8, 2]. The other option is the multiple-component format, in which an exact number is written as an unevaluated sum of ordinary floating-point numbers, each with its own significand and exponent [11, 31, 32, 13].

## 3    The 'NaN' Problem

Hyperbolic space can be modelled with several well-known isomorphic models. Previous works have proposed to optimize over the hyperbolic space with the Poincaré ball model [27] and Lorentz hyperboloid model [28]. In this paper, we propose to optimize within the Poincaré upper-half space model instead, because its properties allow for localization of potential sources of numerical error.

**The Poincaré upper-half space model** is the manifold $(\mathcal{U}^n, g_u)$, where $\mathcal{U}^n = \{\boldsymbol{x} \in \mathbb{R}^n : x_n > 0\}$ is the upper half space of $n$-dimensional Euclidean space. The metric and corresponding distance function are

$$g_u(\boldsymbol{x}) = \frac{g_e}{x_n^2}, \quad d_u(\boldsymbol{x}, \boldsymbol{y}) = \operatorname{arcosh}\left(1 + \frac{\|\boldsymbol{x} - \boldsymbol{y}\|^2}{2x_n y_n}\right)$$

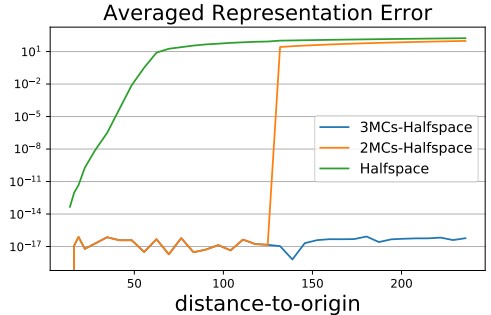
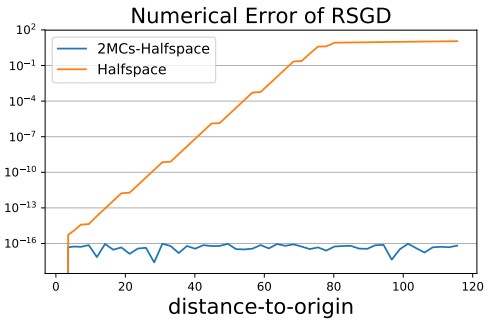

Figure 1: Averaged representation error $d(\boldsymbol{x}, \mathrm{fl}(\boldsymbol{x}))$ as $\boldsymbol{x}$ moves away from the origin in halfspace models.

Figure 2: Numerical error $d(\boldsymbol{y}, \boldsymbol{y}')$ of the RSGD algorithm as $\boldsymbol{x}$ moves away from the origin in halfspace models.

where $g_e$ is the Euclidean metric. Notably, the tangent space of the upper-half space model is the same as the ambient vector space, which makes it easier and numerically stable to derive Riemannian gradients without projection to the tangent space. In the two dimensional case, the Poincaré upper-half space model can be interpreted as a mapping on the complex plane (the "half-plane model").

**Numerical error analysis.** Yu and De Sa [37] shows that in the Lorentz hyperboloid model of the hyperbolic space, the error of representing an exact point $\boldsymbol{x}$ with ordinary floating-point $\mathrm{fl}(\boldsymbol{x})$ is unbounded and proportional to the distance to the origin $d(\boldsymbol{x}, \boldsymbol{O})$. Here, as a baseline to compare our MCF methods to, we show that the same sort of representation error $d(\boldsymbol{x}, \mathrm{fl}(\boldsymbol{x}))$ happens when representing points in the Poincaré upper-half space model with ordinary floating-point numbers.

**Theorem 1** *The representation error of storing a particular point $\boldsymbol{x} \in \mathcal{U}^n$ using floating-points $\mathrm{fl}$ is $\delta_{\mathrm{fl}}(\boldsymbol{x}) = d_u(\boldsymbol{x}, \mathrm{fl}(\boldsymbol{x}))$, and the* worst case representation error *defined as a function of the distance-to-origin $d$ in the Poincaré upper-half space model is*

$$\delta_d := \max_{\boldsymbol{x} \in \mathcal{U}^n, d_u(\boldsymbol{x}, \boldsymbol{O}) \le d} \delta_{\mathrm{fl}}(\boldsymbol{x}) = \operatorname{arcosh}(1 + \epsilon_{machine}^2 \cosh^2(d)).$$

*where $\epsilon_{machine}$ is the machine epsilon of the underlying floating-point arithmetic. This becomes $\delta_d = 2\epsilon_{machine} + o(\epsilon_{machine})$ if $d < \log(1/\epsilon_{machine})$ and $\delta_d = 2d + 2\log(\epsilon_{machine}) + o(\epsilon_{machine}^{-1} \exp(-2d))$ if $d \ge \log(1/\epsilon_{machine})$.*

We plot the averaged representation error and the 'NaN' problem practically in Figure 1, the distance error $d(\boldsymbol{x}, \mathrm{fl}(\boldsymbol{x}))$ when $\boldsymbol{x}$ is represented as $\mathrm{fl}(\boldsymbol{x})$ using floating-point. The error explodes at at distance of around 55 from the origin, which will then potentially cause NaNs in subsequent computations.

**Identify potential numerical errors.** Optimizing $\boldsymbol{x}$ in the Poincaré upper-half space model will typically push $\boldsymbol{x}$ away from the origin, with $x_i, i \ne n$ being large floating-point numbers and $x_n$ being floating-point numbers close to 0. Hence, the subtraction of two big floating-point numbers in $\|\boldsymbol{x} - \boldsymbol{y}\|^2$ of the distance calculation leads to catastrophic cancellation and accounts for a large part of the numerical errors. We can localize this part of the numerical errors easily in simple subtractions in the Poincaré upper-half space model. (For comparison, in the Lorentz model [28], the multiplication in Lorentzian scalar product accounts for the major numerical errors in distance computations.)

**RSGD in the Poincaré upper-half space model.** We are interested in gradient-based optimizations within the Poincaré upper-half space model, hence we provide Riemannian stochastic gradient descent (RSGD) algorithm for different learning algorithms in the hyperbolic space. For each $\boldsymbol{z} \in \mathcal{U}^n$ in the Poincaré upper-half space model, let $f : \mathcal{U}^n \to \mathbb{R}$ be a smooth real-valued loss function over the manifold, denote $\nabla_{\boldsymbol{z}} f$ to be the Euclidean gradient of $f$ w.r.t. $\boldsymbol{z}$. In order to do RSGD in the Poincare upper-half space model, we firstly derive the Riemannian gradient in the tangent space from the Euclidean gradient using the pull-back metric computed as:

$$\operatorname{grad}_{\boldsymbol{z}} f = z_n \nabla_{\boldsymbol{z}} f,$$

then we take the learning rate $\eta$ into consideration, denote $\boldsymbol{v} = -\eta \cdot \text{grad}_{\boldsymbol{z}} f$. Next, we will need the *exponential map* **Exp** in the Poincaré upper-half space model, which takes a tangent vector $\boldsymbol{v}$ in the tangent space to $\boldsymbol{z}'$ on the manifold $\mathcal{U}^n$ via the following arithmetic:

$$z'_i = z_i + \frac{z_n}{\frac{s}{\tanh s} - v_n} \cdot v_i, \qquad i < n \tag{1}$$

$$z'_n = \frac{z_n}{\cosh s - \frac{\sinh s}{s} v_n} \tag{2}$$

where $s = \sqrt{\boldsymbol{v}^T \boldsymbol{v}}$. We also provide a more numerically stable form of this exponential map in the Appendix. As an example, assume an exact point $\boldsymbol{x}$ is given, we do RSGD following the above arithmetic and get the accurate $\boldsymbol{y}$ as:

$$\boldsymbol{y} = \textbf{Exp}_{\boldsymbol{x}}(-\eta \cdot \text{grad}_{\boldsymbol{x}} f).$$

We can derive a $\boldsymbol{y}'$ using ordinary floating-point arithmetic and compute the distance error between the exact $\boldsymbol{y}$ and $\boldsymbol{y}'$. In Figure 2, we show how this error $d_u(\boldsymbol{y}, \boldsymbol{y}')$ varies as the starting point $\boldsymbol{x}$ gradually moves away from the origin. At a distance of around 60, the halfspace RSGD algorithm will be numerically unstable with numerical errors large enough to cause NaNs.

In the Poincaré upper-half space model, the major part of the numerical errors in the exponential map happens in Equation 1, when a big floating-point number is added to a small floating-point gradient number. This lets us localize the numerical errors within a simple addition operation in the Poincaré upper-half space model; in comparison, for the exponential map in the Lorentz model [28], multiple operations cause numerical errors. Hence, we propose to use multiple-component floating-point numbers in the Poincaré upper-half space model to avoid the numerical errors in addition and subtraction, rather than base our methodology in other models like the Lorentz model.

## 4 Multiple-Component Floating-point (MCF)

Currently, most high precision algorithms are developed with one of two approaches: (1) the standard multiple-digit "BigFloat" approach, involving a sequence of digits coupled with a single exponent and (2) the multiple-component format (MCF), which uses unevaluated sums of multiple ordinary floating-point numbers. The multiple-digit approach can represent compactly a much larger range of numbers, whereas the multiple-component approach has an advantage in speed, as it better takes advantage of existing floating-point accelerators. Since speed and scalability is critical for ML applications, MCF is to be preferred here. In this section, we will detail *how* MCF works.

For a machine which supports $p$-bit floating-point number arithmetic, exact arithmetic often produces values that require more than $p$-bits to express and store. In the multi-component floating-point format, each arbitrary precision value $x$ is expressed as an *expansion* [36]:

$$x = x_n + \cdots + x_2 + x_1,$$

where each $x_i$ is called a component of $x$ and is represented by a floating-point number with a $p$-bit significand. As the structure requirements of this expansion definition, all components are ordered by magnitude (from largest $x_n$ to smallest $x_1$). What's more, every two different components $x_i, x_j$ are required to be non-overlapping: the least significant nonzero bit of $x_i$ is more significant than the most significant nonzero bit of $x_j$ or vice versa.

Multiple-component floating-point format allows roundoff error to occur when performing exact arithmetic, then accounts for it afterwards by adding more components. To measure roundoff quickly and correctly, the algorithms presented herein rely on the assumption that ordinary floating-point ops are performed with standard to-nearest rounding. MCF allows the boundaries between components to wander freely, and boundaries are still enforced but can fall at any bit position. For example, the sum of $2^{300} + 2^{-300}$, will be represented by $(2^{300}, 2^{-300})$ in the multiple-component floating-point format. In this sense, the multiple-component approach may store numbers with fewer bits, stored as two words of memory, whereas the multiple-digit approach will use at least 601 bits to store the sum [36].

From Figure 1, we see that the 'NaN' problem occurs at a distance around 50 to the origin, thus for the representation of hyperbolic space, the range of distances is not the main concern, rather, the issue is that when points are far from the origin, a small error in magnitude causes a high numerical

| **Algorithm 1: Two-Sum** | **Algorithm 2: Grow-Expansion** |
|---|---|
| **Input:** $p$-bit floats $a, b$, where $p \geq 3$ | **Input:** $m$ $p$-bits expansion $e$, $p$-bit float $b$ |
| $\quad x \leftarrow \mathbf{fl}(a + b)$ | $\quad$ **initialize** $Q_0 \leftarrow b$ |
| $\quad b_{\text{virtual}} \leftarrow \mathbf{fl}(x - a)$ | $\quad$ **for** $i = 1$ to $m$ **do** |
| $\quad a_{\text{virtual}} \leftarrow \mathbf{fl}(x - b_{\text{virtual}})$ | $\quad\quad (Q_i, h_i) \leftarrow$ **Two-Sum**$(Q_{i-1}, e_i)$ |
| $\quad b_{\text{roundoff}} \leftarrow \mathbf{fl}(b - b_{\text{virtual}})$ | $\quad$ **end for** |
| $\quad a_{\text{roundoff}} \leftarrow \mathbf{fl}(a - a_{\text{virtual}})$ | $\quad h_{m+1} \leftarrow Q_m$ |
| $\quad y \leftarrow \mathbf{fl}(a_{\text{roundoff}} + b_{\text{roundoff}})$ | $\quad$ **Return:** $h$ |
| **Return:** $(x, y)$ | |

error in subsequent computations. Multiple-component floats, which can have very small numerical errors, will thus be better-suited to represent hyperbolic space compared to the multiple-digit format.

Multiple-component floating-point arithmetic can be faster than multiple-digit arithmetic because the latter requires expensive normalization of results to fixed digit positions. MCF can also be implemented to run efficiently on GPUs because their unevaluated-sum structure is amenable to parallelism. This allows learning applications that use MCF to enjoy not only accurate results but also benefit from the hardware acceleration of GPUs.

## 5 Computing with MCF

We provide some basic numerical operations for computing with multiple-component floating-point numbers. To start with, we consider the sum of two $p$-bit floating-point numbers to form a non-overlapping expansion.

**Theorem 2** *[21, 36]. For floating point numbers $a, b$, Alg. 1 produces a non-overlapping expansion $(x, y)$ such that $a + b = x + y$, where $x$ is an approximation to $a + b$ and $y$ is the roundoff error. Particularly, when exact rounding is adopted, the roundoff error $y$ is bounded as*

$$|y| \leq \frac{1}{2}\mathbf{ulp}(\mathrm{fl}(a + b)), \quad |y| \leq \min(|a|, |b|).$$

*where $\mathbf{ulp}(x)$ is the unit of the least precision of $x$.*

This theoretical bound on the roundoff error $y$ serves an important role in computations involving Alg. 1 (most MCF algorithms). Now we can derive an expansion by adding two $p$-bit floating-point values with this algorithm, next, we need to add a single floating-point number to an expansion. Alg. 2 shows how to add a single $p$-bit value to any arbitrary precision value expressed as an expansion.

**Theorem 3** *[36] Let $e = \sum_{i=1}^{m} e_i$ be a non-overlapping $m$ $p(\geq 3)$-bit components expansion, sorted in increasing order, except that any of the $e_i$ may be zero. Let $b$ be a $p(\geq 3)$-bit value, then Alg. 2 produces a non-overlapping $m + 1$ cmponents expansion $h = \sum_{i=1}^{m+1} h_i = e + b$, where the components are also in increasing order, except that any of them may be zero. Notably, $Q_i$ in Alg. 2 can serve as an approximate sum of $b$ and the first $i$ components of $e$.*

As a result of Alg. 2, addition of a value to an $m$-components expansion will grow the expansion and outputs an expansion with $m + 1$ components. However, if there are no constraints on the number of components, then different values will be highly irregular and vary a lot, which will slow down the computations and make the algorithms much more complicated. Hence, in our implementation, rather to grow the number of components without any constraints, we would like to fix the number of components for all expansions we may use. With Alg. 3, we can renormalize an $m + 1$ expansion back to an $m$ components expansion.

Priest [32] proves that if the input expansion of Alg. 3 does not overlap by more than 51 bits, then the algorithm works correctly. This condition holds when ordinary floating-point numbers are used. Table 1 summarizes MCF algorithms used in this paper, some of them detailed in the Appendix. Note that firstly these MCF algorithms can be easily vectorized to apply on high dimensional vectors, secondly, they are linear in the number of components (quadratic for **Scale-Expansion**) and dimension, hence enjoy a great speedup.

Table 1: MCF algorithms used in the paper

| Operation | Function | Example Usage |
|---|---|---|
| **Two-Sum** | sum of two floats | most MCF algorithms |
| **Grow-Expansion** | add a float to an expansion | RSGD, update parameters |
| **Add-Expansion** | sum of two expansions | distance calculation |
| **Scale-Expansion** | multiply a float to an expansion | gradient calculation, RSGD |
| **Renormalize** | reduce # components of an expansion | RSGD, reduce # components |

## 6   Learning using MCF

We solve the 'NaN' problem in the Poincaré upper-half space model using multiple-component floating-point arithmetic in the addition & subtraction computations. As analyzed in Sec. 3, we only need to use $m$-multiple-component floating-point numbers for the first $n-1$ coordinates of the model, denoted as $m$-**xMC-Halfspace**. We will mostly compute with MCF arithmetic (in Table 1) for the addition & subtraction involved in the computations, and leave the rest of computations to ordinary floating-point arithmetic.

For example, we compute the distance between $\boldsymbol{x}, \boldsymbol{y} \in \mathcal{U}^n$ as:

$$d_u(\boldsymbol{x}, \boldsymbol{y}) = \operatorname{arcosh}\left(1 + \frac{\textbf{Add-Expansion}(\boldsymbol{x}, -\boldsymbol{y})_1^2}{2x_n y_n}\right)$$

where we add two expansions $\boldsymbol{x}, -\boldsymbol{y}$ to get the summed expansion and then take the first, largest component as an approximation of the scalar $\|\boldsymbol{x} - \boldsymbol{y}\|$. Furthermore, in Appendix we show how to do RSGD using the multi-component floating-point numbers in the $m$-**xMC-Halfspace** model.

---

**Algorithm 3: Renormalize [32]**

**Input:** $(m+1)$-components expansion $(a_0, a_1, \cdots, a_m)$ in decreasing order.
**initialize** $s \leftarrow a_m, k \leftarrow 0$
**for** $i = m$ to $1$ **do**
  $(s, t_i) \leftarrow$ **Two-Sum**$(a_{i-1}, s)$
**end for**
**for** $i = 1$ to $m$ **do**
  $(s, e) \leftarrow$ **Two-Sum**$(s, t_i)$
  **if** $e \neq 0$ **then**
    $b_k \leftarrow s$
    $s \leftarrow e$
    $k \leftarrow k + 1$
  **end if**
**end for**
**Return:** $(b_0, b_1, \cdots, b_{m-1})$

---

When we compute with the introduced multi-components floating-point numbers in the model, we show theoretically the worse case numerical representation error in Theorem 4.

**Theorem 4** *The worst case representation error of storing an exact point $\boldsymbol{x} \in \mathcal{U}^n$ with $m$-multi-component floating-point expansion $(\boldsymbol{x}^{(m)}, \cdots, \boldsymbol{x}^{(2)}, \boldsymbol{x}^{(1)})$ defined as a function of the distance-to-origin $d$ is*

$$\delta_d = \operatorname{arcosh}\left(1 + \epsilon_{machine}^2 + \frac{\epsilon_{machine}^{2m}(1 + \epsilon_{machine})}{2^{2m}}\cosh^2(d)\right),$$

*where $\epsilon_{machine}$ is the machine epsilon. This becomes $\delta_d = 2\epsilon_{machine} + o(\epsilon_{machine})$ if $d < m\log(1/\epsilon_{machine})$ and $\delta_d = 2d + 2m\log(\epsilon_{machine}/2) + o(\epsilon_{machine}^{-2m}\exp(-2d))$ if $d \geq m\log(1/\epsilon_{machine})$.*

Table 2: Datasets Information

| Datasets | Nodes | Edges |
|---|---|---|
| WordNet[12] | 74374 | 75834 |
| ↳ Nouns | 82115 | 769130 |
| ↳ Verbs | 13542 | 35079 |
| ↳ Mammals | 1181 | 6541 |
| Gr-QC[22] | 4158 | 13422 |

Table 3: Embedding performances on Mammals

| Model | MAP | MR |
|---|---|---|
| Halfspace | 92.07%±1.01% | 1.600±0.79 |
| 2-xMC-Halfspace | **93.85**%±0.57% | **1.430**±0.16 |
| Lorentz | 85.75%±2.30% | 1.857±4.78 |
| L-Tiling | 91.49%±1.42% | 1.645±1.28 |
| H-Tiling | 91.60%±1.60% | 1.559±0.44 |

| DIMENSION | MODELS | WORDNET NOUNS | | WORDNET VERBS | | GR-QC | |
|---|---|---|---|---|---|---|---|
| | | MAP | MR | MAP | MR | MAP | MR |
| 2 | LORENTZ | 15.61±0.37% | 55.59±0.53 | 56.76±0.52% | 4.18±0.07 | 55.93±0.32% | 71.81±1.24 |
| | L-TILING | 23.39±0.31% | 34.22±1.03 | 63.08±0.31% | 3.28±0.03 | 56.07±0.21% | 72.53±2.08 |
| | H-TILING | 23.34±0.19% | 34.30±0.49 | 64.26±0.40% | 3.23±0.05 | 55.95±0.30% | **71.78**±1.02 |
| | HALFSPACE | 23.31±0.17% | 34.81±0.26 | 63.65±0.32% | 3.36±0.07 | 55.60±0.35% | 74.29±1.32 |
| | 2-XMC-HALFSPACE | **24.42**±0.11% | **34.01**±0.17 | **64.88**±0.25% | **3.18**±0.05 | **56.35**±0.38% | 71.81±1.12 |
| 5 | LORENTZ | 74.82±0.38% | 7.528±0.23 | 89.72±0.04% | 1.52±0.00 | 71.16±0.33% | 33.66±1.23 |
| | H-TILING | 74.90±0.08% | 7.529±0.12 | 90.13±0.08% | 1.46±0.00 | 70.89±0.15% | 32.99±0.52 |
| | HALFSPACE | 74.89±0.14% | 7.388±0.08 | 89.99±0.21% | 1.51±0.01 | 71.10±0.19% | 32.29±0.75 |
| | 2-XMC-HALFSPACE | **75.05**±0.12% | **7.377**±0.12 | **90.25**±0.09% | **1.41**±0.01 | **71.21**±0.24% | **31.89**±0.71 |
| 10 | LORENTZ | 78.43±0.42% | 6.148±0.15 | **90.71**±0.13% | 1.41±0.00 | **72.84**±0.13% | 28.53±0.27 |
| | H-TILING | 78.40±0.22% | 6.229±0.13 | 90.62±0.02% | 1.40±0.00 | 72.57±0.09% | **28.50**±0.35 |
| | HALFSPACE | 78.39±0.12% | 6.227±0.21 | **90.71**±0.11% | 1.39±0.02 | 72.66±0.11% | 28.56±0.24 |
| | 2-XMC-HALFSPACE | **78.46**±0.08% | **6.136**±0.14 | **90.71**±0.04% | **1.39**±0.00 | 72.69±0.09% | 28.57±0.32 |

Table 4: Learning experiments on different datasets. Results are averaged over 5 runs and reported in mean+std style. **Bold** numbers show the best performance among all models.

Hence, there's a region around the origin whose radius is proportional to the number of components $m$, where we can be confident there are no numerical issues. This is the scaling we'd expect, and is in line with the results from [34]. In Figure 1, we plot the averaged representation error $d(\boldsymbol{x}, \mathbf{MCF}(\boldsymbol{x}))$ when $\boldsymbol{x}$ is represented using 2-component floating-points. The error will not explode until a distance of 125 from the origin simply by using 2 components, compared to the distance of 55 for vanilla halfspace.

Particularly, we also show the numerical errors of the adapted RSGD algorithm in the 2-**xMC-Halfspace** model in Figure 2, with way smaller error compared to the increasing numerical errors in Poincaré upper-half space model using ordinary floating numbers. Practitioners can choose the number of components according to the desired precision requirements, yet many applications would get full benefits from using merely a small multiple of (such as two or three) components.

As a matter of fact, MCF can be more generally adopted in many operations, in all hyperbolic models and different generic loss functions for better performance. Hence we also implement the model with all coordinates represented with MCF, denoted as $m$-**MC-Halfspace**. We defer the discussion of this model to Appendix since it is more sophisticated by making use of the **Scale-Expansion** algorithm.

## 7 Experiments

Firstly proposed in Nickel and Kiela [27], hyperbolic embedding was used to embed data with hierarchical structure such as taxonomies, producing results that outperform Euclidean embeddings. Motivated by this, we conduct embedding experiments on datasets including the Gr-Qc and commonly used WordNet Nouns, Verbs and Mammals dataset whose statistics are shown in Table 2. We are interested in two aspects of the proposed MCF-based models in practice: firstly the capacity of the models, and secondly the running speed of model training and optimization.

**Reconstruction.** We focus on the tasks which can show the capacity of the models. Given an observed transitive closure $\mathcal{D} = \{(u, v)\}$ of all objects, we embed all objects into hyperbolic space and then reconstruct it from the embedding. Hence, the error from the reconstructed embedding to the ground truth will be a reasonable measure for the capacity of the model and reflect how the underlying floating-point arithmetic may affect the performance. Specifically, we learn embeddings of the transitive closure $\mathcal{D}$ such that related objects are close in the embedding space. To do this, we aim to minimize the loss introduced in [27]:

$$\mathcal{L}(\Theta) = \sum_{(u,v) \in \mathcal{D}} \log \frac{e^{-d(\boldsymbol{u},\boldsymbol{v})}}{\sum_{\boldsymbol{v}' \in \mathcal{N}(u)} e^{-d(\boldsymbol{u},\boldsymbol{v}')}},$$

where $\mathcal{N}(u) = \{v \mid (u, v) \notin \mathcal{D}\} \cup \{u\}$ is the set of negative examples for $u$ (including $u$). We do negative sampling randomly per positive example during training for different datasets.

We implemented the upper-half space model and **xMC-Halfspace** models trained with RSGD in our experiments, as for baselines, we consider the current state-of-the-art $L$-tiling models, $H$-tiling models [37] trained with RSGD and the Lorentz model [28]. Models were trained using ordinary

| Dataset | Models | Batchsize=32 | | | Batchsize=64 | | | Batchsize=128 | | |
|---|---|---|---|---|---|---|---|---|---|---|
| | | 2D | 5D | 10D | 2D | 5D | 10D | 2D | 5D | 10D |
| Wordnet Nouns | Lorentz | 57.37 | 60.00 | 53.42 | 27.38 | 27.28 | 26.74 | 9.37 | 9.08 | 9.14 |
| | Halfspace | 118.7 | 119.9 | 119.7 | 40.2 | 40.3 | 39.3 | 17.4 | 17.4 | 20.4 |
| | $L$-tiling (CPU) | 205 | **203** | 203 | 185 | 185 | 182 | 161 | 158 | 158 |
| | $H$-tiling (CPU) | **146** | **188** | 238 | 107 | 156 | 212 | 93 | 129 | 177 |
| | 2-xMC-Halfspace | 272 | 284 | 296 | 118 | 136 | **121** | **54** | **58** | **68** |
| Wordnet Verbs | Lorentz | 3.21 | 3.19 | 3.25 | 0.93 | 0.93 | 0.96 | 0.42 | 0.41 | 0.41 |
| | Halfspace | 5.65 | 5.68 | 5.71 | 1.69 | 1.66 | 1.66 | 0.78 | 0.75 | 0.77 |
| | $H$-tiling (CPU) | **7.46** | **8.87** | **10.76** | **5.56** | 7.85 | 8.54 | 5.06 | 6.63 | 8.59 |
| | 2-xMC-Halfspace | 12.38 | 12.14 | 12.83 | 6.19 | **5.86** | 6.42 | 1.80 | **1.84** | 2.09 |
| Gr-QC | Halfspace | 4.29 | 4.36 | 4.31 | 2.13 | 1.91 | 1.88 | 0.59 | 0.63 | 0.67 |
| | Lorentz | 2.45 | 2.61 | 2.44 | 0.67 | 0.76 | 0.67 | 0.36 | 0.36 | 0.36 |
| | $H$-tiling (CPU) | **5.99** | **7.22** | **10.12** | **4.39** | 6.45 | 8.12 | 4.16 | 5.41 | 7.32 |
| | 2-xMC-Halfspace | 9.52 | 9.96 | 10.41 | 5.42 | 5.57 | 5.94 | **1.46** | 1.63 | **1.79** |

Table 5: Training time (seconds per epoch) for different models on different datasets. Results are averaged over 10 epochs. Performances of low precision models (Lorentz&Halfspace) are separated from that of high precision models in the table.

float64 with the same hyper-parameters starting from same initializations of parameters, see Appendix for more implementation details. To evaluate the quality of the embeddings, we make use of the standard graph embedding metrics in [5, 26]. For an observed relationship $(u, v)$, we rank the distance $d(\boldsymbol{u}, \boldsymbol{v})$ among the set $\{d(u, v')|(u, v') \in \mathcal{D}\}$, then we evaluate the ranking on all objects in the dataset and record the mean rank (MR, smaller⇒better) as well as the mean average precision (MAP, larger⇒better) of the ranking.

We firstly evaluate different 2-dimensional embeddings on the light-scale Mammals dataset. As shown in Table 3, we choose 2 components for the MCF-based models. The proposed upper-half space model already outperforms all previously proposed baseline models. In particular, the **xMC-Halfspace** model further improves the embedding performance and achieves a 2.25% MAP improvement on Mammals compared to the best baseline models.

We present the large scale experiments on other three datasets in Table 4. These results show that MCF-based models generally perform better than baseline models in various dimensions with respect to MAP and MR, particularly in 2 dimensional hyperbolic space, since higher dimension can alleviate the 'NaN' problem to some degree. On 2 dimensional embeddings of the largest WordNet Nouns dataset, MCF-based models gain as large as a 1.03% MAP improvement compared to the $L$-tiling model and a 1.11% MAP improvement compared to the upper-half space model.

The limitation of the work is that in some cases of embedding problems that stay close to the origin, MCF will give no improvement over plain half space because it was already numerically accurate enough, also our method will eventually break down at very large distances due to overflow or underflow of the floating point exponent.

**Training time cost.** Running the model on GPUs confers important training advantage for modern machine learning tasks. In order to provide a comprehensive comparison result, we train most models in our experiments on GPUs including MCF-based half space models, upper-half space model and the Lorentz model. As for the $L$-tiling and $H$-tiling model, since big-integer matrix multiplications are not supported in GPUs, we train both models with CPUs. For all timing results within this section, we use GPU: GeForce RTX 2080 Ti and CPU: Intel(R) Xeon(R) Gold 6240 @2.60GHZ.

Note that negative sampling size and batch size can make a great difference to the timing performance over CPUs and GPUs. Herein, we fix the negative sampling size to be 50 and only vary the batchsize in 32, 64, 128. Readers may question the performances of the model as the batchsize changes, we note that with a simple scale of the learning rate, one can achieve comparable results for different batchsize, detailed in the Appendix. We report the training time (seconds per epoch) for different models on three different datasets in Table 5. We can conclude that:

- On large-scale datasets or large batchsize (e.g. 128) training, MCF-based models (high precision) can be trained efficiently, with timing close to that of models using ordinary floating-point.

- MCF-based models (high precision) can run on GPUs, enjoying a great training speedup compared to other high precision models such as the $L$-tiling and $H$-tiling models which can only run over CPUs. As a matter of fact, on WordNet Nouns, relatively it takes MCF-based models around 2/3 less time and saves 30.27 hours to finish 1000 epochs training (batchsize = 128), compared to the H-Tiling model with best MAP performance.

More experiment details are in the Appendix. We will release our code in PyTorch publicly for reproducibility, and in hopes that practitioners can use MCF to reliably compute in hyperbolic space.

## 8 Conclusion

In this work, we proposed upper-half space model of hyperbolic space and presented new multi-component floating-point representations. Our approach avoids the numerical errors inherent to popular models of hyperbolic space, such as the Lorentz model, while supporting efficient computations on GPUs. This allows our MCF approach to effectively trade off computational cost and numerical accuracy for learning over hyperbolic space.

## Acknowledgments and Disclosure of Funding

This work is supported by NSF IIS-2008102. This work was supported in part by a gift from SambaNova Systems, Inc. The authors would like to thank Aaron Lou and anonymous reviewers from NeurIPS 2021 for providing valuable feedbacks on earlier versions of this paper.

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
