# Appendix: Representing Hyperbolic Space Accurately using Multi-Component Floats

**Tao Yu**
Department of Computer Science
Cornell University
tyu@cs.cornell.edu

**Christopher De Sa**
Department of Computer Science
Cornell University
cdesa@cs.cornell.edu

## 1 Proofs of Theorems

Here we first provide the proof of the worst case representation error in the Poincaré upper-half space model.

**Theorem 1.** *The representation error of storing a particular point $\boldsymbol{x} \in \mathcal{U}^n$ using floating-points fl is $\delta_{\mathrm{fl}}(\boldsymbol{x}) = d_u(\boldsymbol{x}, \mathrm{fl}(\boldsymbol{x}))$, and the* worst case representation error *defined as a function of the distance-to-origin $d$ in the Poincaré upper-half space model is*

$$\delta_d := \max_{\boldsymbol{x} \in \mathcal{U}^n, d_u(\boldsymbol{x},\boldsymbol{O}) \leq d} \delta_{\mathrm{fl}}(\boldsymbol{x}) = \mathrm{arcosh}(1 + \epsilon_{machine}^2 \cosh^2(d)).$$

*where $\epsilon_{machine}$ is the machine epsilon of the underlying floating-point arithmetic. This becomes $\delta_d = 2\epsilon_{machine} + o(\epsilon_{machine})$ if $d < \log(1/\epsilon_{machine})$ and $\delta_d = 2d + 2\log(\epsilon_{machine}) + o(\epsilon_{machine}^{-1} \exp(-2d))$ if $d \geq \log(1/\epsilon_{machine})$.*

*Proof.* Consider the error $\delta_{fl}(\boldsymbol{x})$ first as

$$\begin{aligned}
\delta_{fl}(\boldsymbol{x}) &= \mathrm{arcosh}(1 + \frac{\sum_{i=1}^n \epsilon_i^2 x_i^2}{2x_n \mathrm{fl}(x_n)}) \\
&= \mathrm{arcosh}(1 + \frac{\sum_{i=1}^n \epsilon_i^2 x_i^2}{2(1+\epsilon_n)x_n^2}) \\
&\leq \mathrm{arcosh}(1 + \frac{\epsilon_{machine}^2 \|x\|^2}{2(1+\epsilon_n)x_n^2}) \\
&= \mathrm{arcosh}(1 + \epsilon_{machine}^2 \cdot \frac{\|x\|^2}{2x_n^2} + o(\epsilon_{machine}^3)).
\end{aligned}$$

On the other hand, note that

$$\begin{aligned}
\cosh d(\boldsymbol{x}, \boldsymbol{O}) &= 1 + \frac{\sum_{i=1}^{n-1} x_i^2 + (x_n - 1)^2}{2x_n} \\
&= 1 + \frac{\|x\|^2 + 1 - 2x_n}{2x_n} \\
&= \frac{\|x\|^2 + 1}{2x_n}.
\end{aligned}$$

35th Conference on Neural Information Processing Systems (NeurIPS 2021).

Hence, $2x_n \cosh d = \|x\|^2 + 1$, where $d = d(\boldsymbol{x}, \boldsymbol{O})$, then we have

$$\delta_{fl}(\boldsymbol{x}) = \text{arcosh}(1 + \epsilon_{machine}^2 \cdot \frac{\|x\|^2}{2x_n^2} + o(\epsilon_{machine}^3)).$$

$$\leq \text{arcosh}(1 + \epsilon_{machine}^2 \cdot \frac{2x_n \cosh d - 1}{2x_n^2} + o(\epsilon_{machine}^3))$$

$$\leq \text{arcosh}(1 + \epsilon_{machine}^2 \cdot (\frac{\cosh d}{x_n} - \frac{1}{2x_n^2}) + o(\epsilon_{machine}^3))$$

$$= \text{arcosh}(1 + \epsilon_{machine}^2 \cdot (t \cosh d - \frac{1}{2}t^2) + o(\epsilon_{machine}^3))$$

where $t = \frac{1}{x_n}$, note that $t \cosh d - \frac{1}{2}t^2$ attains maximum $\frac{1}{2}\cosh^2 d$ at $t = \cosh d$, therefore, $\delta_{fl}(\boldsymbol{x}) \leq \text{arcosh}(1 + \frac{\epsilon_{machine}^2 \cosh^2 d}{2} + o(\epsilon_{machine}^3))$, then we can derive

$$\delta_d := \max_{\boldsymbol{x} \in \mathcal{U}^n, d_u(\boldsymbol{x}, \boldsymbol{O}) \leq d} \delta_{fl}(\boldsymbol{x}) = \text{arcosh}(1 + \epsilon_{machine}^2 \cosh^2 d).$$

With taylor expansion when $d < \log(1/\epsilon_{machine})$ and $d \geq \log(1/\epsilon_{machine})$, the conclusion follows. $\qquad\square$

Here we provide the proof of the worst case representation error in the $m$-**xMC-Halfspace** model, where MCF is only adopted for the first $n-1$ axes.

**Theorem 4.** *The worst case representation error of storing an exact point $\boldsymbol{x} \in \mathcal{U}^n$ with $m$-multi-component floating-point expansion $(\boldsymbol{x}^{(m)}, \cdots, \boldsymbol{x}^{(2)}, \boldsymbol{x}^{(1)})$ (increasing order) defined as a function of the distance-to-origin $d$ is*

$$\delta_d = \text{arcosh}\left(1 + \epsilon_{machine}^2 + \frac{\epsilon_{machine}^{2m}(1 + \epsilon_{machine})}{2^{2m}} \cosh^2(d)\right),$$

*where $\epsilon_{machine}$ is the machine epsilon. This becomes $\delta_d = 2\epsilon_{machine} + o(\epsilon_{machine})$ if $d < m\log(1/\epsilon_{machine})$ and $\delta_d = 2d + 2m\log(\epsilon_{machine}/2) + o(\epsilon_{machine}^{-2m}\exp(-2d))$ if $d \geq m\log(1/\epsilon_{machine})$.*

*Proof.* Firstly, note due to the construction of multi-component floats, the error caused by the floating point only exists at the smallest component, since MCF accounts for the errors in previous components afterwards by adding more components. Due to the non-overlapping property of the MCF, we have $x^{(m+1)} \leq \frac{1}{2}\text{ulp}(x^{(m)}) = \frac{1}{2}\epsilon_{machine}x^{(m)}$, hence, we can sequentially derive that $x^{(m)} \leq 2^{-(m-1)}\epsilon_{machine}^{m-1}x^{(1)}$, then similarly, consider the error $\delta_{fl}(\boldsymbol{x})$ as

$$\delta_{fl}(\boldsymbol{x}) = \text{arcosh}(1 + \frac{\sum_{i=1}^{n-1} \epsilon_i^2(x_i^{(m)})^2 + \epsilon_n^2 x_n^2}{2x_n \text{fl}(x_n)})$$

$$\leq \text{arcosh}(1 + \frac{\sum_{i=1}^{n-1} \epsilon_i^2(2^{-(m-1)}\epsilon_{machine}^{m-1}x_i^{(1)})^2 + \epsilon_n^2 x_n^2}{2(1 + \epsilon_n)x_n^2})$$

$$\leq \text{arcosh}(1 + \frac{2^{-2(m-1)}\epsilon_{machine}^{2m}\|x\|^2 + (1 - 2^{-2(m-1)})\epsilon_{machine}^2 x_n^2}{2(1 + \epsilon_n)x_n^2})$$

$$\leq \text{arcosh}(1 + 2^{-2(m-1)}\epsilon_{machine}^{2m}(1 + \epsilon_{machine}) \cdot \frac{\|x\|^2}{2x_n^2} + \epsilon_{machine}^2(1 + \epsilon_{machine})/2).$$

also we have $2x_n \cosh d = \|x\|^2 + 1$, where $d = d(\boldsymbol{x}, \boldsymbol{O})$, then we have

$$\frac{\|x\|^2}{2x_n^2} = \frac{2x_n \cosh d - 1}{2x_n^2}$$

$$= \frac{\cosh d}{x_n} - \frac{1}{2x_n^2}$$

$$= t \cosh d - \frac{1}{2}t^2$$

where $t = \frac{1}{x_n}$, note that this attains maximum $\frac{1}{2}\cosh^2 d$ at $t = \cosh d$, therefore, $\delta_{fl}(\boldsymbol{x}) \leq \text{arcosh}(1 + 2^{-2(m-1)}\epsilon_{machine}^{2m}(1 + \epsilon_{machine}) \cdot \cosh^2 d + \epsilon_{machine}^2(1 + \epsilon_{machine})/2)$, then we can derive

$$\delta_d := \max_{\boldsymbol{x} \in \mathcal{U}^n, d_u(\boldsymbol{x}, \boldsymbol{O}) \leq d} \delta_{fl}(\boldsymbol{x}) = \text{arcosh}(1 + \epsilon_{machine}^2 + \frac{\epsilon_{machine}^{2m}(1 + \epsilon_{machine}) \cdot \cosh^2 d}{2^{2m}}).$$

With taylor expansion when $d < m\log(1/\epsilon_{machine})$ and $d \geq m\log(1/\epsilon_{machine})$, the conclusion follows. $\qquad\square$

## 2  Gradient Computations

Here we show how to compute gradients in the halfspace model: assume two points $\boldsymbol{u}, \boldsymbol{v} \in \mathbb{R}^d$, we will compute the gradient $\boldsymbol{g_u} \in \mathbb{R}^d$ of the hyperbolic distance w.r.t. $\boldsymbol{u}$ as follows:

$$x = \frac{\|\boldsymbol{u} - \boldsymbol{v}\|^2}{2u_d v_d}, \quad z = \sqrt{x(x+2)}, \quad \boldsymbol{y} = \frac{\boldsymbol{u} - \boldsymbol{v}}{u_d v_d},$$

$$(\boldsymbol{g_u})_{1:d-1} = \frac{1}{z}\boldsymbol{y}_{1:d-1},$$

$$(\boldsymbol{g_u})_d = \frac{1}{z}(\boldsymbol{y}_d - \frac{x}{u_d}).$$

We can either choose to compute this gradients in ordinary floating-point arithmetic, or compute them with the adapted MCF arithmetic using the provided MCF algorithms in the $m$-**xMC-Halfspace** model. In our implementaions, we compute this gradients with the adapted MCF arithmetic.

## 3  Numerical stable form of Exp

Here we offer a numerical stable form of the aforementioned exponential map **Exp**. Firstly, for the first equation regarding the $x$-axes, i.e.,

$$z_i' = z_i + \frac{z_n}{\frac{s}{\tanh s} - v_n} \cdot v_i.$$

Note that if $v_n \geq 0$, then the subtraction in the denominator of $\frac{s}{\tanh s}$ (close to 1) to $v_n$ (close to 0) is the major part to the numerical error, therefore, we'd like to avoid this subtraction with a different computation but in the same arithmetic as follows:

$$
\begin{aligned}
z_i' &= z_i + \frac{z_n}{\frac{s}{\tanh s} - v_n} \cdot v_i \\
&= z_i + \frac{z_n}{s\coth s - v_n} \cdot v_i \\
&= z_i + \frac{s\coth s + v_n}{s^2\coth^2 s - v_n^2} \cdot z_n \cdot v_i \\
&= z_i + \frac{s\coth s + v_n}{s^2\csc^2 s + s^2 - v_n^2} \cdot z_n \cdot v_i \\
&= z_i + \frac{s\coth s + v_n}{s^2\csc^2 s + r^2} \cdot z_n \cdot v_i,
\end{aligned}
$$

where $s = \sqrt{\boldsymbol{v}^T \boldsymbol{v}}$, and $r^2 = \sum_{i=1}^{n-1} v_i^2$. In this way, we can avoid the numerical error caused by the subtraction when $v_n \geq 0$, note that if $v_n < 0$, then the 'subtraction' is actually an addition, hence we will keep the original formula.

For the second equation regarding the $y$-axis, i.e.,

$$z_n' = \frac{z_n}{\cosh s - \frac{\sinh s}{s} v_n}.$$

Again note that if $v_n \geq 0$, the subtraction in the denominator of $\cosh s$ (close to 1) to $\frac{\sinh s}{s} v_n$ (close to 0) is the major part to the numerical error, here we also provide a different computation but with the same arithmetic to avoid the subtraction when $v_n \geq 0$ as follows:

$$
\begin{aligned}
z'_n &= \frac{z_n}{\cosh s - \frac{\sinh s}{s} v_n} \\
&= \frac{s \cdot z_n}{s \cosh s - v_n \sinh s} \\
&= \frac{s \cosh s + v_n \sinh s}{s^2 \cosh^2 s - v_n^2 \sinh^2 s} \cdot s \cdot z_n \\
&= \frac{s \cosh s + v_n \sinh s}{r^2 \cosh^2 s + v_n^2 \cosh^2 s - v_n^2 \sinh^2 s} \cdot s \cdot z_n \\
&= \frac{s \cosh s + v_n \sinh s}{r^2 \cosh^2 s + v_n^2} \cdot s \cdot z_n.
\end{aligned}
$$

Similarly, we avoid the numerical error caused by the subtraction when $v_n \geq 0$, and if $v_n < 0$, then the 'subtraction' is actually an addition, hence we will keep the original formula.

---

**Algorithm 1: Add-Expansion**, modified from [4]

> **Input:** $m$-components expansions $(a_1, \cdots, a_m)$ and $(b_1, \cdots, b_m)$, both in decreasing order.
> **initialize** $e \leftarrow 0$
> **for** $i = 1$ to $m$ **do**
>   $(h_p, e_1) \leftarrow$ **Two-Sum**$(a_i, b_i)$
>   $(h_i, e_2) \leftarrow$ **Two-Sum**$(h_p, e)$
>   $e \leftarrow \mathbf{fl}(e_1 + e_2)$
> **end for**
> $h_{m+1} \leftarrow e$
> **Return:** $(h_1, \cdots, h_m, h_{m+1})$

---

## 4 More Algorithms Operating MCF

We intend to provide more algorithms to compute MCF expansions in this section, to begin with, the sum of two expansions in Alg. 1 **Add-Expansion**, different from its version firstly proposed in [4], we modify the algorithm to output an expansion with $m + 1$.

Next, we aim to offer algorithms for multiplication of an expansion to a single $p$-bit floating-point number here. To begin with, we need an algorithm for multiplications between two $p$-bit floating-point numbers to form a non-overlapping expansion, termed as **Two-Prod**. We firstly show the following Lemma 1 for the purpose, particularly designed for 53-bit IEEE double precision floating point numbers.

---

**Algorithm 2: Split**

> **Input:** 53-bit double precision floats $a$
>   $t \leftarrow \mathbf{fl}((2^{27} + 1) \cdot a)$
>   $a_{hi} \leftarrow \mathbf{fl}(t - \mathbf{fl}(t - a))$
>   $a_{lo} \leftarrow \mathbf{fl}(a - a_{hi})$
> **Return:** $(a_{hi}, a_{lo})$

---

**Lemma 1.** *[1] Alg. 2 **Split** splits a 53-bit IEEE double precision floating point number into $a_{hi}$ and $a_{lo}$, each with 26 bits of significand, such that $a = a_{hi} + a_{lo}$. $a_{hi}$ contains the first 26 bits, while $a_{lo}$ contains the lower 26 bits. Note that this algorithm can be easily generated to any $p$-bit floating-point number [4].*

With this, we show how to multiply two $p$-bit floating-point numbers to get an non-overlapping expansion in Alg. 3 **Two-Prod**.

**Algorithm 3: Two-Prod**

**Input:** double precision floats $a, b$

$\quad p \leftarrow \mathbf{fl}(a \cdot b)$
$\quad (a_{hi}, a_{lo}) \leftarrow \mathbf{Split}(a)$
$\quad (b_{hi}, b_{lo}) \leftarrow \mathbf{Split}(b)$
$\quad err_1 \leftarrow \mathbf{fl}(x - \mathbf{fl}(a_{hi} \cdot b_{hi}))$
$\quad err_2 \leftarrow \mathbf{fl}(err_1 - \mathbf{fl}(a_{lo} \cdot b_{hi}))$
$\quad err_3 \leftarrow \mathbf{fl}(err_2 - \mathbf{fl}(a_{hi} \cdot b_{lo}))$
$\quad y \leftarrow \mathbf{fl}(\mathbf{fl}(a_{lo} \cdot b_{lo}) - err_3)$

**Return:** $(x, y)$

---

**Theorem 2.** *[1, 4] Alg. 3 computes $p = \mathbf{fl}(a \cdot b)$ and corresponding roundoff error $e = \mathbf{err}(a \cdot b)$.*

Herein, we provide the multiplication algorithm of an expansion to a single $p$-bit floating-point number in Alg. 4 **Scale-Expansion**. Note that both Alg. 4 **Scale-Expansion** and Alg. 1 **Add-Expansion** grows the expansion only one more to be $m + 1$ components, hence, we will need to apply the **Renormalize** algorithm to reduce the number of components.

---

**Algorithm 4: Scale-Expansion**, modified from [4]

**Input:** $m$-components expansion $(a_1, \cdots, a_m)$ in decreasing order, $p$-bit float $b$.
**initialize** $e \leftarrow 0$
**for** $i = 1$ to $m$ **do**
$\quad (h_p, e_1) \leftarrow \mathbf{Two\text{-}Prod}(a_i, b)$
$\quad (h_i, e_2) \leftarrow \mathbf{Two\text{-}Sum}(h_p, e)$
$\quad e \leftarrow \mathbf{fl}(e_1 + e_2)$
**end for**
$h_{m+1} \leftarrow e$
**Return:** $(h_1, \cdots, h_m, h_{m+1})$

---

## 5 RSGD & MCs-Halfspace Model

We provide the RSGD algorithm adapted in the $m$-**xMCs-Halfspace** model using the provided MCF algorithms in Alg. 5:

---

**Algorithm 5:** RSGD in the $m$-**xMCs-Halfspace** model

**Require:** Objective function $f$
**Require:** $\boldsymbol{z} \in \mathcal{U}^n$, Epochs $T$, and learning rate $\eta$
$\quad$ **for** $t = 0$ to $T - 1$ **do**
$\qquad \mathrm{grad}_{\boldsymbol{z}} f \Leftarrow z_n \nabla_{\boldsymbol{z}} f, \triangleright$ Riemannian gradient
$\qquad \boldsymbol{v} = -\eta \cdot \mathrm{grad}_{\boldsymbol{z}} f, \triangleright$ learning rate
$\qquad s \Leftarrow \sqrt{\boldsymbol{v}^T \boldsymbol{v}}$
$\qquad \mathrm{grad}_{\boldsymbol{z}_{1:n-1}} \Leftarrow \frac{z_n}{\frac{s}{\tanh s} - v_n} \cdot v_i, \triangleright$ gradient of $x$-axis values
$\qquad w \Leftarrow \mathbf{Grow\text{-}Expansion}(z_{1:n-1}, \mathrm{grad}_{\boldsymbol{z}_{1:n-1}})$
$\qquad z'_{1:n-1} \Leftarrow \mathbf{Renormalize}(w), \triangleright$ Update $x$-axis values
$\qquad z'_n = \frac{z_n}{\cosh s - \frac{\sinh s}{s} v_n}, \triangleright$ Update $y$-axis values
$\quad$ **end for**
**output** $\boldsymbol{z}'$

---

As mentioned in the main body of the paper, we can appy MCF on all coordinates to get the **m-MC-Halfspace** model. The distance computations within this model are consistent to the **m-xMC-Halfspace** model, while the key difference is the usage of the Alg. 4 **Scale-Expansion** in the

**m-MC-Halfspace** model, since the last coordinate of the model is involved mostly in multiplications. More importantly, we show in Alg. 6 how to do RSGD in the $m$**-MCs-Halfspace** model.

---

**Algorithm 6:** RSGD in the $m$**-MCs-Halfspace** model

---

**Require:** Objective function $f$
**Require:** $\boldsymbol{z} \in \mathcal{U}^n$, Epochs $T$, and learning rate $\eta$
   **for** $t = 0$ to $T - 1$ **do**
      $\text{grad}_{\boldsymbol{z}} \, f \Leftarrow z_n \nabla_{\boldsymbol{z}} f, \triangleright$ Riemannian gradient
      $\boldsymbol{v} = -\eta \cdot \text{grad}_{\boldsymbol{z}} f, \triangleright$ learning rate
      $s \Leftarrow \sqrt{\boldsymbol{v}^T \boldsymbol{v}}$
      $\text{grad}_{\boldsymbol{z}_{1:n-1}} \Leftarrow \frac{z_n}{\frac{s}{\tanh s} - v_n} \cdot v_i, \triangleright$ gradient of $x$-axis values
      $w_x \Leftarrow$ **Grow-Expansion**$(z_{1:n-1}, \text{grad}_{\boldsymbol{z}_{1:n-1}})$
      $z'_{1:n-1} \Leftarrow$ **Renormalize**$(w_x), \triangleright$ Update $x$-axis values
      $w_y \Leftarrow$ **Scale-Expansion**$(z_n, \frac{1}{\cosh s - \frac{\sinh s}{s} v_n})$
      $z'_n \Leftarrow$ **Renormalize**$(w_y), \triangleright$ Update $y$-axis values
   **end for**
**output** $\boldsymbol{z}'$

---

In this way, for the addition & subtraction occurring in the exponential map, between a potentially large floating-point coordinate number to a small floating-point gradient is done in MCF arithmetic. Notice that we only adopt MCF arithmetic in part of the distance, gradient and exponential map computations, and leave the rest of computations computed in ordinary floating-point arithmetic.

## 6 Experiment details

We conducted our experiments based on the implementation of [5], with all learning experiments in PyTorch based on ordinary float64. For the initialization of all models, we drawn randomly from the uniform distribution $U(-1e-5, 1e-5)$. Particularly, for the initialized embedding of the upper-half space model and MCF-based models, the last axis is initialized with $1 + U(-1e-5, 1e-5)$, since the corresponding origin in halfspace model is $(0, \cdots, 0, 1)$.

We train embeddings of different dataset for 1000 epochs except for the largest WordNet-Nouns dataset with 500 epochs. At the start of the training, we train models with an initial "burn-in" phase firstly proposed in [2], which helps find a good initial angular layout and a good resulting embedding, simply by using a reduced learning rate $\eta/100$.

For most of the hyper-parameters in our experiment, we adopt the recommended values from the implementations of [5, 2, 3], the negative sampling size is 50 in the experiments. We vary batchsize within $\{32, 64, 128\}$ and use grid search to find the optimal learning rate in each case.

We mention an interesting tuning result here, take the training of the halfspace model over the WordNet Mammal for example, we varies the learning rates for different batchsize as shown in Table. 1. We found that, if trained with a larger batchsize, when the learning rate is adjusted (increased) properly, the embedding performance of the converged model with a large batchsize can nearly match the best performance of the converged model with a smaller batchsize. Similar phenomenon was observed for the rest dataset in different dimensions for different models. Hence, we can safely choose batchsize=128 in our main experiments for its running time advantage, with a learning rate 5.0. We provide the code together with the parameters of our implementation in the supplementary material.

| BATCHSIZE | LR | MAP (%) | MR |
|---|---|---|---|
| 32 | 0.3 | 72.96 | 2.53 |
| | 1.0 | 87.65 | 1.99 |
| | 2.0 | 93.65 | 1.43 |
| | 3.0 | 90.95 | 1.81 |
| 64 | 0.3 | 29.34 | 18.45 |
| | 1.0 | 84.97 | 1.78 |
| | 2.0 | 89.90 | 1.76 |
| | 3.0 | 92.29 | 1.57 |
| | 4.0 | 92.36 | 1.66 |
| 128 | 1.0 | 61.79 | 3.98 |
| | 2.0 | 81.12 | 2.28 |
| | 3.0 | 87.45 | 1.89 |
| | 4.0 | 91.27 | 1.55 |
| | 5.0 | 92.07 | 1.50 |
| | 6.0 | 92.32 | 1.56 |
| | 20.0 | 87.80 | 2.79 |

Table 1: Embedding performances of the halfspace model on the wordnet Mammal dataset with different hyperparameter setting.