# OpenReview forum: "Representing Hyperbolic Space Accurately using Multi-Component Floats"
_NeurIPS.cc/2021/Conference — NeurIPS 2021 Poster_

### Official Review · Reviewer_gMed · 2021-07-13

**Rating:** 6
**Confidence:** 3

**Summary:**

To solve the 'NaN' problem in hyperbolic embeddings, the paper proposes to learn the hyperbolic embeddings in the Poincare upper-half space model using multi-component floating-point (MCF). Theoretically, the paper proves that numerical errors can be reduced to any degree by simply increasing the number of components of MCF. In experiments, it shows the proposed MCF can have better results while not compromising much on efficiency.

**Limitations And Societal Impact:**

Yes, the authors addressed the limitations. The potential negative societal impact is not discussed since they thought there are none, which I agree.

**Main Review:**

The paper focuses on the numerical issues in the hyperbolic embeddings. The task and experimental settings closely follow the previous work on tiling-based models, but the proposed method is novel since it uses the Poincare upper-half space to learn the hyperbolic embeddings and it utilizes MCF to alleviate the numerical errors. The theoretical contribution is solid and clearly presented. However, the improvements in empirical evaluation is very small. The proposed method is quite close with H-Tiling. The gap is smaller than std sometimes.

*Post-rebuttal*: I appreciate the response of the authors. I am keeping my score.

**Time Spent Reviewing:**

2.5

---

> ### Author Response · Authors · 2021-08-08
> **Response to Review**
>
> Hi, we thank the reviewer for considering our work as novel and with solid theoretical contribution. For the concern that our empirical improvement over the tiling-based model is small, we want to note that the goal of our paper is to compute accurately and efficiently on GPUs. As shown in [37], tiling-based models have bounded representation errors everywhere; however they can not run efficiently on GPUs due to the big-integer matrix computations involved. Our goal is for our method to have performances as close as possible to the tiling-based models (potentially better with more components), while computing much faster (on GPUs), as shown in Table 5, our proposed models are *2.6 times faster than the H-Tiling model under the 128 batchsize case.

---

### Official Review · Reviewer_FLfq · 2021-07-16

**Rating:** 7
**Confidence:** 2

**Summary:**

Solving "the NaN problem" in the Poincaré upper-half space model using multiple-component floating-point arithmetic.

**Limitations And Societal Impact:**

The authors adequately addressed the limitations and potential negative societal impact of their work

**Main Review:**

In this paper, the authors propose a simple, feasible-on-GPUs, and easy-to-understand solution for numerically accurate learning on Poincaré upper-half space model using multiple-component floating-point arithmetic in the addition & subtraction computations.

The work is technically sound and well organized.
But (1) Poincar\'e Ball model and Lorentz model are more popular in hyperbolic embedding, how about using MCF on these models? (2) Many works use SGD or Adam for optimization, how about using these optimizers?

Totally, the work is new and important to the research community.

**Time Spent Reviewing:**

4 Hours

---

> ### Author Response · Authors · 2021-08-08
> **Response to Review**
>
> Hi, we thank the reviewer for considering our work as important to the community and well organized.
> 1. MCFs can definitely be used with Poincaré and Lorentz models. However, as mentioned in Line 158, in the Ball and Lorentz models the errors occur mostly in multiplications, in contrast to the upper-half space model, where the numerical errors are caused by subtractions. As a result, to use MCFs with Poincaré or Lorentz models, we would need to use algorithms for multiplications of two expansions, which are more complex and introduce significantly more computational slowdown. In comparison, only MCFs algorithms for addition & subtractions are needed for the upper-half space model. Furthermore, the upper-half space model also performs no worse than the other two models in the considered tasks.
> 2. In our implementation, we use the Riemannian SGD for optimization, which is the natural analog of SGD on curved surfaces. One can also choose to apply SGD naively for optimization: however as shown in [37], SGD performs worse than Riemannian SGD. We will consider using Adam or Riemannian Adam optimizers in our following work, but RSGD already performs very well on our tasks.

---

### Official Review · Reviewer_72MY · 2021-07-16

**Rating:** 7
**Confidence:** 3

**Summary:**

This paper proposes a model for hyperbolic space using multi-component floating point in the Poincare upper-half space model which can be supported on GPU and with small numerical error. Experimental results show good training time performance.

**Limitations And Societal Impact:**

The author does mention that one of the limitation is that some  embeddings problems may not benefit from the proposed method.

Some questions and suggestions:
1. Can the proposed method be possibly benchmarked for its performance on model trained using it? Such as model accuracy.
2. While the authors does provide a thorough introduction on multi-component fp representations, a figure illustrating it directly would be better.
3. I would suggest to summarise or have a table for training time also in relative improvements. As it seems the saving time in training time can be huge compared to 3% improvements in MAP, which is also presented in relative comparison.
4. Can we also have some experiments on how does the proposed method perform on problems that it may not benefit from?

**Main Review:**

This paper offers a new model to support hyperbolic space multi-component floating-point representations. The proposed method offers good trade-off between numerical error magnitude and hardware cost. This paper is well organised for readers to comprehend the problems and the proposed solution. The authors provide enough background discussing the related models and their limitations in computational precision and existing framework support.  The proposed algorithms is clearly demonstrated with sound mathematical analysis and supported with experimental results comparing training time on several graph-related data sets.

**Time Spent Reviewing:**

3

---

> ### Author Response · Authors · 2021-08-08
> **Response to Review**
>
> Hi, we thank the reviewer for regarding our work to be meaningful and well organised.
> 1. While it’s totally doable to benchmark our method for downstream tasks such as link prediction, we believe this is outside the scope. Our goal in the paper is to come up with an accurate and computationally efficient representation, which will then be evaluated for its representational accuracy compared with ordinary floating-point for the same task.
> 2. We will include a figure to better illustrate the MCFs and summarize the relative training time improvement to Table 5.
> 3. In Table 4, we can see that on the GR-QC dataset, the proposed method performs similar to the halfspace baseline model: this is what we would expect based on our theory for tasks where the dataset is smaller and embeddings are close to the origin.

---

### Decision · Program_Chairs · 2021-09-27

**Decision:**

Accept (Poster)

**Comment:**

The paper proposes a new approach to numerically represent hyperbolic space with high precision using multi-component floats. Insufficient precision is known to cause problems for hyperbolic representation learning and this paper proposes an interesting method to alleviate this issue. The contributions are somewhat incremental as there exists closely related prior work (also in terms of organization of the paper) on improving numerical accuracy for hyperbolic embeddings by Yu and De Sa (2019). However, an important advantage of the proposed method is that it allows for easy GPU-training what significantly expands it applicability compared to prior work. Reviewers highlighted also the sound mathematical analysis and empirical evaluation in the manuscript. After rebuttal, all reviewers supported therefore acceptance of the paper. When preparing the camera ready version of the manuscript, please take the feedback and comment from the reviewers into account to improve the paper.